# Post-Tensioned Hollow-Core Concrete Slabs with Unbonded Tendons for Truck Scale Platforms: Design Assumptions and Tests

**DOI:** 10.3390/ma17164154

**Published:** 2024-08-22

**Authors:** Rafał Stanisław Szydłowski, Barbara Łabuzek, Łukasz Bednarski

**Affiliations:** 1Faculty of Civil Engineering, Cracow University of Technology, 31-155 Cracow, Poland; 2TCE Structural Design & Consulting, 31-409 Cracow, Poland; blabuzek@tce-building.com; 3Faculty of Mechanical Engineering and Robotics, AGH University of Science and Technology, 30-059 Cracow, Poland; lukaszb@agh.edu.pl

**Keywords:** hollow-core slabs, prestressed concrete, truck scale, unbonded tendons

## Abstract

At Cracow University of Technology, attempts were made to develop national truck scale platforms with a capacity of 60 tons, made from prestressed concrete. For this work, we designed slabs partially prestressed with unbonded tendons featuring a cross-section of 1.00 × 0.28 m and a span of 5.94 m. To reduce the weight of the slabs, four channels made from commonly used ø110 × 2.2 mm PVC pipes were used. In this way, we created post-tensioned hollow-core slabs. Due to the unpredictable behavior of slabs operating in a cracked state under a repetitive load, two slabs were subjected to cyclic loads amounting to 1,000,000 cycles with different load values. This paper presents the basic design principles and design details of the slabs, as well as the methodology and results of the research conducted. Lastly, we provide appropriate conclusions directed at further optimizing the slabs.

## 1. Introduction

The European market offers a wide range of truck scales, usually with a nominal load capacity of 6 to 90 tons [1,2]. The basic requirements for truck scale platforms include high durability, stiffness, and light weight. Scales with a working range up to 20 tons are used most frequently as portable steel platforms. Scales with larger nominal capacities are produced as recessed or overground structures, usually made from steel, reinforced and prestressed concrete, or steel–concrete (steel bearing beams with a filling RC panel). Unfortunately, these three requirements are mutually exclusive in any constructional solution. Truck scales with steel platforms are lighter but remain prone to corrosion and require additional resources to maintain their condition during their lifetime. Reinforced concrete and steel–concrete slabs are cheaper to maintain and have higher durability than steel platforms. The main disadvantage of RC slabs is their high weight, which increases the costs for transporting such slabs. For example, the weight of a single module of a reinforced concrete platform with a capacity of 60 tons and a length of 6.0 m can be over 10 tons, while the weight of a whole platform with a length of 18.0 m can be 32 tons. The solution that best fulfills all requirements is post-tensioned concrete. This technology allows the construction of slender panels (light in weight) with high stiffness and good durability.

At Cracow University of Technology, we attempted to design lightweight platforms made from prestressed concrete for vehicle scales with a working range of up to 60 tons. The slabs were designed to be partially prestressed (operating as cracked) using unbonded tendons (monostrands) due to the low weight of the structures relative to the service load and established prestressing limits.

The problems in designing partially prestressed elements have been a subject of interest among many researchers. Since the 1960s, numerous studies and research reports, conducted primarily in the US and, to a lesser extent, in Europe, have been published in this area (Albelsey [3], Lin [4], Moustafa [5], and Naaman et al. [6,7,8]). The reactions of partially prestressed elements subjected to cyclic loading have also been studied, albeit to a much lesser extent [9,10]. Undoubtedly, the least explored issue is the behavior of elements prestressed by unbonded tendons under cyclic loads. Post-tensioned columns with unbonded tendons have been subjected to many tests under cyclic loading [11,12], as have post-tensioned walls with unbonded tendons [13]. However, a study on partially prestressed slender slabs with unbonded tendons under cyclic loads has not yet been published. The present study seeks to fill this research gap.

Given the calculation difficulties in assessing the behavior of slabs during the expected period of use at the Research Laboratory of Cracow University of Technology, we constructed the two slabs with the design and technology typically used for platforms. The slabs were tested under cyclic loads. The number of cycles was matched with the lifetime of a platform and the intensity of its use, and the value of the load was matched with the characteristics of weighed vehicles. The paper presents the basic assumptions, design problems, applied design solutions, and results of the conducted tests, along with comments and conclusions.

## 2. Materials and Methods

### 2.1. General Characteristics of the Platform

For this study, we designed a platform with dimensions of 18.0 × 3.0 m and constructed a platform composed of three simply supported spans, each with a span of 6.0 m. Another objective was to reduce the total weight of the platform to 24 tons based on the maximum load capacity of a standard 20 ft container. This value is optimal from the perspective of transportation economics. Therefore, we decided to design the platform using two parallel concrete tracks for the wheels with a cross-section of 0.9 × 0.28 m, connected with steel I-beams. The plan for the platform is presented in Figure 1, while the cross-section is given in Figure 2.

After analyzing different variants of vehicles and possible loads, we ultimately adopted a design using three axles, each spaced 1.25 m apart, with the load on each axle amounting to 115 kN (Figure 3). One of the key objectives of this study was to develop a light weight and relatively low-cost platform. Initially, we intended to produce slabs with pretensioned concrete technology, which could provide reproducible (prefabricated) and cheap (mass-produced) platforms. After the preliminary static and economic analyses were performed, the proposed solution was found to be unacceptable for two reasons. First, pretensioned concrete elements working under such harsh environmental conditions should operate without cracks. This feature requires a sufficiently large amount of prestressing to reduce the high bending stresses caused by a vehicle load. However, opposite stresses (prestressing) are introduced in situations without service loads (the initial design of prestressed structures). This situation does not allow the use of prestressing with large eccentricity due to the possibility of cracking of the upper surface, which is not desirable in an aggressive environment. Reducing the high tensile stresses caused by bending requires a large amount of prestressing when one contends with the presence of a large overhanging tendon and cannot apply a large eccentric load. The second reason was the excessive cost of commissioning the production line to create pretensioned concrete elements completely different from existing ones.

Considering the above, we adopted post-tensioning technology with unbonded tendons. The cross-section of the platform was assumed to include two elements with the necessary width to allow free entry of the car wheels; these elements were connected with steel beams. Figure 1 shows a plan of the platform, while Figure 2 and Figure 4 show transverse and longitudinal cross-sections. We then constructed a platform transverse cross-section consisting of two slabs with a cross-section of 0.90 × 0.28 m and axial spacing of 2.10 m (total width of the platform: 3.00 m). The longitudinal arrangement provides three types of panels: S-1, S-2, and S-3 slabs. These slabs are supported by hot-rolled steel angle brackets on 8 force gauges. All slabs have the same dimensions as the concrete element and differ only in the geometry of the steel supporting elements. The adopted assumptions enable the creation of a platform 12.0 m in length (abandoning the central slab S-2) or 24.0 m (doubling the size of slab S-2).

### 2.2. Characteristics of Prestressed Slabs

Originally, we assumed dimensions of 0.9 × 0.28 mm for the full cross-section of the slab. A preliminary analysis of the calculations showed that designing a slab capable of working in a non-cracked state would be extremely difficult due to the small share of the slab’s own weight in the total load. Implementing crack resistance required the use of 16 15.5 mm strands in each slab. Therefore, we chose to design partially prestressed elements containing 4 unbonded tendons (Figure 5) in each slab with an overhang of 50 mm and a characteristic strength of 1860 MPa. The initial force in each tendon was set to 200 kN. The slab was reinforced near the bottom surface with 12 ribbed bars of 16 mm in diameter. Transverse reinforcement used 6 mm diameter doubled strips with spacing every 250 mm. In addition, 6 longitudinal bars with diameters of 8 mm were used near the top surface. The slabs were made of class C40/50 concrete (according to EC2) based on basalt aggregate and Portland cement. The slabs were prestressed no earlier than 14 days after concreting.

After producing a few platforms, a width of 0.9 m proved to be too narrow for practical application of the scales and was, therefore, increased to 1.0 m. Further platforms were constructed using panels with cross-sections of 1.0 × 0.28 m. These panels were also too heavy. The weight of the slab significantly exceeded 4 tons, while the weight of the whole platform exceeded a value of 24 tons. Consequently, the platform exceeded the load of standard trucks and could not be transported as a whole, which significantly increased the cost of transport. We then found an easy and low-cost solution, which involved reducing the weight of the slabs. Since weakening a cross-section at the edges strongly reduces the moment of inertia, which is very important in a prestressed cross-section, the introduction of empty spaces at the bottom edge of the slab was undesirable. Consequently, we decided to use empty cores inside the section of the slab (Figure 5). For this purpose, standard PVC pipes ø110 × 2.2 mm were used. Four cores with a length of 5.0 m were introduced, with the corresponding locations shown in Figure 5. Each of the cores decreased the weight of the slab by approximately 240 kg. Thus, the whole slab became 960 kg lighter, while the platform became 5760 kg lighter.

Table 1 lists the predicted values of bending moments and stresses caused by individual load components in the central cross-section at the top and bottom surfaces alongside the values of predicted stresses after prestressing. The stresses shortly after prestressing were −0.5 MPa at the upper surface and +5.9 MPa at the lower surface. However, in a service situation, the theoretical stresses (under a vehicle load and after time-dependent prestress losses) in the cross-section would be 14.0 MPa at the upper surface and −9.1 MPa at the lower surface.

The crack width defined by the standard [14] is 0.21 mm, the spacing of cracks is 0.31 m, and stress in the reinforcing steel (calculated from the state of cross-section decompression) is 245.9 MPa.

Figure 6a shows a view of the first slab during post-tensioning, while Figure 6b presents a view of the completed platform at its location of use. During the evaluation of the final form, we produced 3 platforms with a dense cross-section of 0.9 × 0.28 mm and 2 with a dense cross-section of 1.0 × 0.28 mm. The following platforms were used in the final version shown in Figure 5.

### 2.3. Test Program

#### 2.3.1. Samples and Test Stand

The test involved two identical slab samples. The samples were made in laboratory of the Cracow University of Technology. All materials provided in the project and utilized for mass production were used to produce the slabs. Figure 7 shows the slabs before concreting. Table 2 lists the test results for the concrete after 28 days. The mechanical characteristics were determined using cylindrical samples 150 mm in diameter and 300 mm in height (compressive strength, modulus of elasticity, and splitting tensile strength) and 0.15 × 0.15 × 0.60 m beam samples (modulus of rupture).

Figure 8 shows the scheme of the test stand along with the location of the measuring gauges. Figure 9 shows a view of the stand. The frontal angle bracket support was rotated by 180° relative to the designed slabs. The bracket’s geometry and affixing scheme were, however, preserved. In addition, a two-point load was used for the slab. We selected 0.85 m spacing for the loading rails to achieve the same bending moment as that of the three-point load caused by three axles (2 × 90 kN instead of 3 × 60 kN) under a two-point load.

The following measuring transducers were installed:Three strain gauges with a length of 5 mm installed on three reinforcement bars in a cross-section through a crack;Eight strain gauges with a base of 75 mm on the two side surfaces of the slab (four on each side) using the arrangement shown in Figure 8;Five displacement transducers to measure the deflection and curvature of the slab;Two transducers to measure crack width (on both side surfaces—Figure 8b);Four prestress force transducers (on each cable, alternating with two at each end of the slab).

#### 2.3.2. Program for Loading

Slabs No. 1 and No. 2 were prestressed 89 and 112 days after concreting. The cables were tensioned under a force of 200 kN with the same equipment and technology used to prestress mass-produced slabs. Figure 10 shows the strain distribution at the cross-section of the center as a result of post-tensioning based on the strain values measured by the strain gauges on the side surfaces. The strain values at the edges of the cross-section were estimated by extending the strain curves. The same figure shows the average values of forces in the tendons after anchoring (177 kN). The values in the two slabs are similar (equal to 6 mm) and differ slightly from the design value (4.4%) calculated for slippage of the wedges in real-life anchorages (169.6 kN). Direct conversion of the strain into stress is always burdened by errors due to differences in the modulus of elasticity for small samples of concrete compared to that for full-scale structures. For this reason, stresses determined using strains should be regarded as only a rough estimation. For slab No. 1, the stresses were calculated as −0.81 MPa for the upper edge and 8.3 MPa for the lower edge. The same values for slab No. 2 were calculated as −0.64 MPa and 5.9 MPa, respectively. Meanwhile, the predicted values were −2.6 and 8.0 MPa (Table 1). Notably, slab No. 2 presented significantly smaller strain values, despite a similar prestressing force. This result was confirmed by camber to be 1.98 mm for slab No. 1 and 1.8 mm for slab No. 2. This result could be due to differences in the modulus of elasticity or prestressing geometry.

The slabs designed for truck scale platforms are used for weighing vehicles in depots with materials, quarries, factories, mines, landfills, etc. The scales are used with very different intensity levels depending on the location of use. The load intensity can range from a few vehicles a day to several hundred. In the designed program, we assumed a conventional number of 250 load cycles per day and a 10-year lifetime for the slab. The total number of load cycles was set to 912,500, with the slab subjected to 1,000,000 load cycles. The designed load of one slab on the platform was 180 kN for 20% vehicle overloading, whereas the design load with vehicle overloading was set to 150 kN. Therefore, the value of the load cyclically varied from 20 to 150 kN. The lower value was due to technical conditions (ensuring the stability of the test stand at a given time). The load frequency was 1.7 Hz, and loading was stopped approximately every 150,000 cycles. After a 30 min break, all the monitored values were registered, and the slab was subjected to a static load of 180 kN (control load) in one cycle before being completely unloaded. During this cycle, all monitored parameters were continuously measured. After completing cyclic loading, the first tested slab was subjected to a load until it was destroyed.

Based on the very positive results of slab No. 1 throughout the loading process, the cyclic load amplitude value for slab No. 2 was increased to 40–180 kN, and the frequency was reduced to 1.5 Hz.

## 3. Test Results

Figure 11 illustrates the deflection and crack width development at the beginning (after 1000 cycles) and end (after a million cycles for slab No. 1 and 888,000 cycles for slab No. 2) of the repetitive load test. For slab No. 1, when the load changed from 20 to 150 kN, deflection initially changed from 4.4 to 20.4 mm (continuous dark blue line) and from 5.7 to 22.9 mm at the end (dotted dark blue line). The crack width changed from 0.01 to 0.11 mm at the start and from 0.01 to 0.12 mm at the end. For slab No. 2, when subjected to a load change within the limits of 40 and 180 kN, deflection oscillated between 5.3 and 26.2 mm at the start (continuous red line) and between 8.4 and 31.2 mm at the end (dotted red line). The crack width changed from 0.025 to 0.18 mm at the start and from 0.03 to 0.21 mm at the end. The results indicate that the deflection and crack width grew alongside an increase in the load cycles, especially for slab No. 2.

The first slab tested under a cyclically variable load of 150 kN successfully handled 1,000,000 load cycles. The changes in the behavior of this slab after increasing the number of cycles is discussed later in this section considering changes in the measured values, together with the results for slab No. 2. After completion, the program was loaded again—this time, until the slab was destroyed. The course of selected parameters up to the destruction test is shown in Figure 12. The extension value of the press piston was used as a measure of deformation. Figure 12a shows the dependence of the force generated by the press and piston displacement. The greatest load capacity was reached by the slab when the piston extended by 110 mm, with a value of 364 kN. In that position, the load capacity had already begun to fall dramatically. The rapid collapse of the load–displacement dependence occurred under a displacement area of 60 mm. The sudden loss of rigidity at that point was also confirmed by a sudden increase in crack width (Figure 12c). Figure 12b shows the increased average force in prestressing strands, which rose from 176.7 to 219.5 kN. Figure 12d illustrates the increased stress values in the reinforcement bars, calculated based on the average strain of strain gauges on the bars. These stresses were determined based on an elasticity modulus value equal to 202.3 GPa using a bar tensile test. In the displacement area of 60 mm, the strain gauges were damaged due to a sudden loss of stiffness. The largest registered stress was 336 MPa, and the average width of the crack was 0.40 mm.

The attempted destruction of the slab showed that the programmed upper value for a cyclically variable load of 150 kN is 41%, while the load amplitude alone is 36% of the breaking load. We generally assumed that a variable cyclic load less than 0.4 times that of the breaking load would not lead to RC element failure due to fatigue. Considering the positive test results for the first slab (small permanent changes in monitored values—see the following figures), we decided to increase the upper limit and amplitude of cyclic loading when loading slab No. 2. The upper value of the force amounted to 180 kN, while the bottom value was increased to 40 kN. The upper load value was then 49%, and the amplitude alone was 38% of the breaking load. This slab unexpectedly failed after approximately 980,000 cycles, a few hours before the scheduled completion of the loading process. The damage was caused by fatigue and a sudden increase in deflection, breaking 9 out of 12 reinforced bars (Figure 13). The two external bars at the first edge and the corner bar at the second edge remained undamaged. Excessive deflection resulted in immediate arrest of the hydraulic cylinder, causing a cyclic load.

Figure 14 shows an arrangement of registered cracks on one of the side surfaces of slab No. 2. Here, the cracks formed immediately upon loading the slab with a force of 180 kN are marked in red. Blue indicates the propagation of cracks after 137,000 load cycles. Further cyclic loading revealed no coverage growth or new cracks. The average spacing of cracks in the cross-section between concentrated forces amounted to 130 mm, with widths ranging from 0.1 to 0.2 mm.

The wait time was approximately 30 min after the cyclic loadings were stopped, after which the residual deflection of the slab was registered, as shown in Figure 15a. Due to prestressing, the 2 mm camber of slab No. 1 and 1.8 mm camber of slab No. 2 were permanently eliminated during the initial loading of the slabs. Residual deflections increased with the number of load cycles. For slab No. 1 (under a cyclic load of 150 kN) and slab No. 2 (under a cyclic load of 180 kN), the residual deflection was 1.1 mm after 1 million cycles and 1.9 mm after 888 thousand cycles.

Figure 15b shows the results of increasing deflections when a control load of 180 kN was applied. The results show an increase in value accompanied by an increase in the number of cycles, especially for slab No. 2. Here, the deflection under a control load was much higher shortly after post-tensioning because post-tensioning reduced the effect of the camber.

The significant increase observed in the deflection of slab No. 2, between 763 and 888 thousand cycles, was produced by a significant decrease in the force of one of the strands (Figure 16), which is discussed in the analysis of prestressing forces. Excluding this case from further analysis, the increase in deflections was 21.3 mm for slab No. 1 and 23.9 mm for slab No. 2, constituting 1/278 and 1/248 of the span, respectively. Thus, we can conclude that the limit value was almost achieved for the two slabs based on a limit value of L/250.

Figure 16 shows the course of prestressing force depending on the load applied at different stages of the cyclic load for slab No. 2. For both slabs, a decrease in the initial and final prestressing force with the number of load cycles was observed in the process of loading. This observation highlights the declining role of prestressing and the increasing role of ordinary reinforcement as the number of cycles increases. Indeed, this result was confirmed by the stress increases in ordinary steel (Figure 17b). The drastic decrease in prestressing force after 888 thousand cycles was caused by a decrease in the force of one strand. The initial value of the force in the cable decreased from 182 kN to 152 kN, i.e., 16%. This decrease was likely caused by a fatigue interruption in one of the seven wires in the strand (a 14% loss in the strand cross-sectional area).

Figure 17 shows the changes in the width of cracks and stresses in reinforcing steel with an increase in the number of cycles. In both cases, an upward trend can be observed. The width of the crack remained at a safe level throughout the entire range of loading. In slab No. 1, the width fluctuated between 0.14 and 0.15 mm, while in slab No. 2, the width fluctuated between 0.18 and 0.19 mm. Notably, the width of the crack was smaller than the calculated value of 0.21 mm. The average spacing of cracks was 130 mm—more than two-times lower than the theoretical value of 310 mm. The stresses in ordinary steel oscillated around a calculated value of 245.9 MPa for slab No 1 and clearly increased with the number of cycles for slab No. 2.

## 4. Conclusions

The results of the conducted tests indicated the following:At a nominal load of 150 kN, the slab resisted 1,000,000 load cycles without any significant increase in the values indicating worsening serviceability conditions or durability (crack width, residual deflection, and deflection increase under service load).At a cyclic load of 180 kN, the slab failed at the end of the programmed load (after 980,000 cycles). However, this result considers a 20% overload of the vehicle, and the probability of such a load occurring in nature is equal to zero based on the number of cycles established in the tests.For the more strongly loaded slab No. 2, we observed a decrease in the role of prestressing as the number of cycles increased (Figure 16). Cross-sectional tensile forces were taken up by the ordinary reinforcement (Figure 17b).The obtained results were not compared with those from other tests, as similar studies have not yet been published. There is also a lack of calculation procedures for analytically determining the studied parameters.Further optimization to decrease the thickness of the slabs or the amount of reinforcement and prestressing may be unachievable as the slab already reached its deflection limits, and stresses in ordinary reinforcements achieved a high level of almost 290 MPa. Further savings could be achieved in the weights of slabs and the amount of concrete by increasing the number of channels.

This paper presents the development results for the first Polish slabs for truck scales made from prestressed concrete with unbonded tendons, as well as the results of laboratory tests under repetitive loads. These slabs were invented, tested, and implemented for mass production by the authors of this paper. The process of shaping these slabs into their final form required approximately two years of work. At that time, three versions of the slabs were created. Notably, there is another European patent on a slab featuring a similar load capacity and thickness of 350 mm. However, the present slab is 280 mm thick and was made with a core version, which significantly reduced its weight.

Our results show that these slabs were properly designed and should work successfully for many years with proper maintenance and the avoidance of unforeseen factors. The period of use, however, will depend on the aggressiveness of the environment causing corrosion of steel and concrete.

Notably, although the present results are unique, they cannot be generalized in any way due to the unexplored nature of the observed phenomena. Plates with similar parameters to those used in this study may behave similarly. However, with changes in any parameter (crack width, load, or stress level in the reinforcement), their behavior will be completely different (as observed in the differences between slabs No. 1 and 2).

Ultimately, our positive experience with the analysis and maintenance of slabs under the studied conditions of use contributed to the development of a novel truck scale with a capacity of 30 tons. This scale features a single-span concrete platform 3.00 × 8.00 m in size and 240 mm thick, prestressed with 10 unbonded tendons.

## Figures and Tables

**Figure 1 materials-17-04154-f001:**
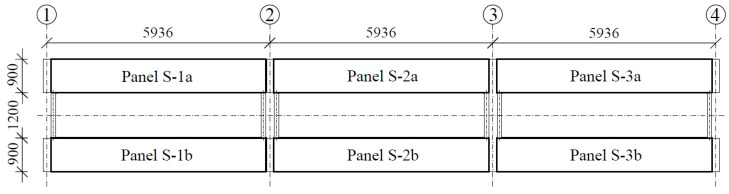
Plan of the platform.

**Figure 2 materials-17-04154-f002:**
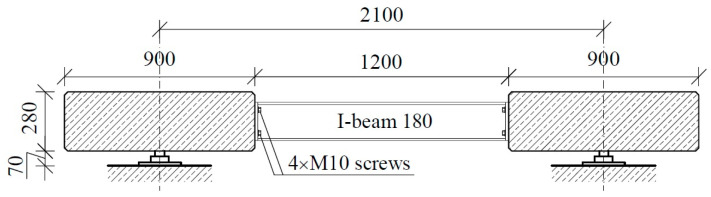
Platform cross-section.

**Figure 3 materials-17-04154-f003:**
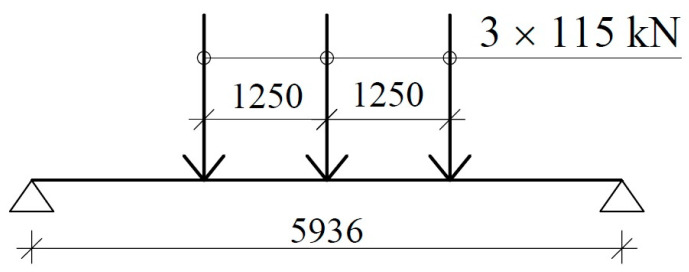
Design load scheme of the selected span.

**Figure 4 materials-17-04154-f004:**
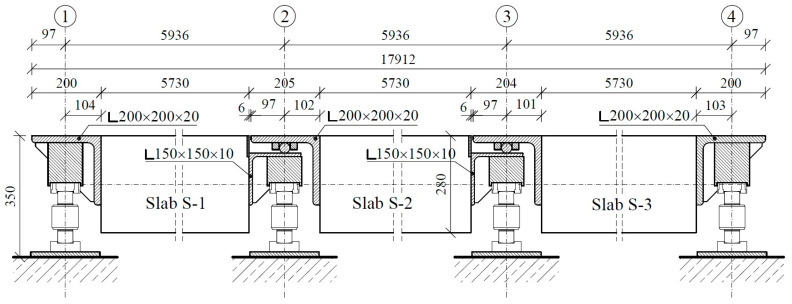
Platform longitudinal cross-section.

**Figure 5 materials-17-04154-f005:**
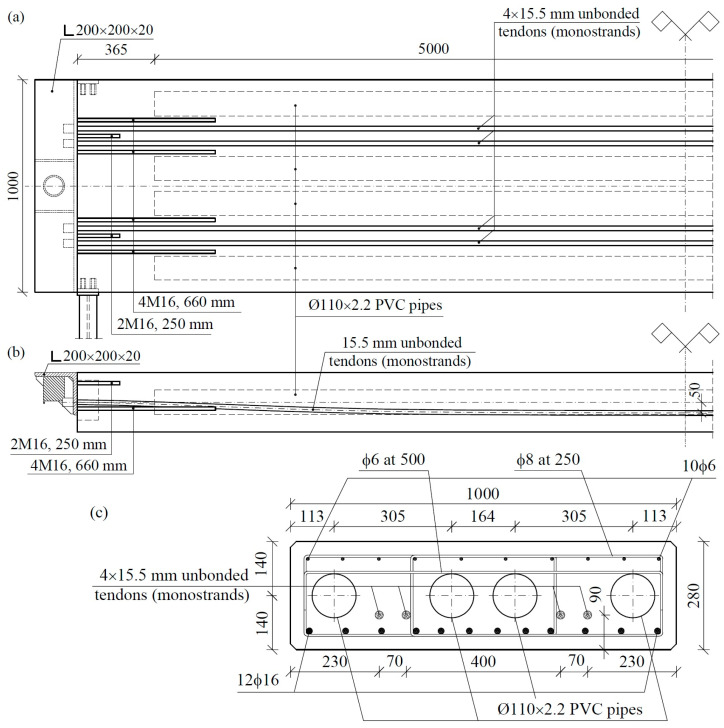
Post-tensioned truck scale slab: horizontal plan (**a**), longitudinal cross-section (**b**), middle-span cross-section (**c**).

**Figure 6 materials-17-04154-f006:**
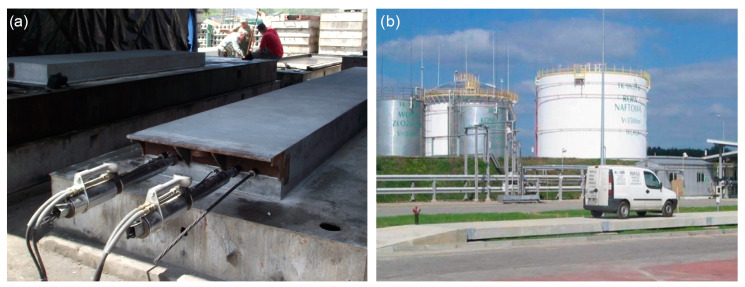
View of the first slab during post-tensioning (**a**); view of the completed platform (**b**).

**Figure 7 materials-17-04154-f007:**
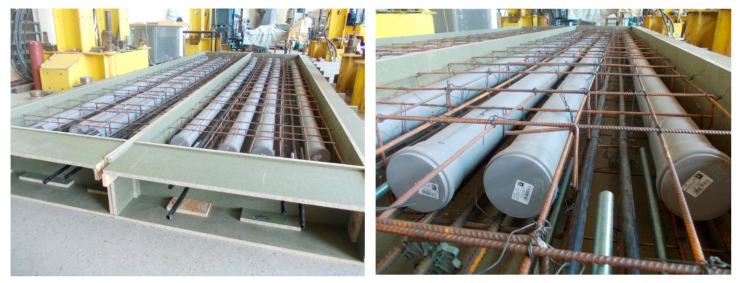
The sample slabs before concreting.

**Figure 8 materials-17-04154-f008:**
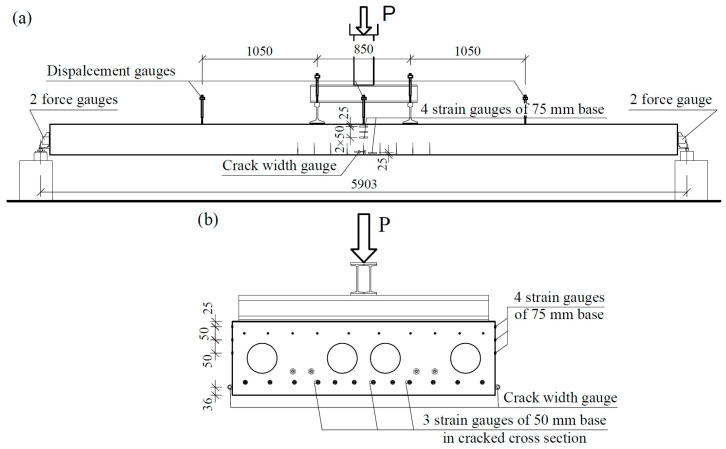
The scheme of the test stand with the locations of the measuring gauges: side view (**a**) and slab cross-section (**b**).

**Figure 9 materials-17-04154-f009:**
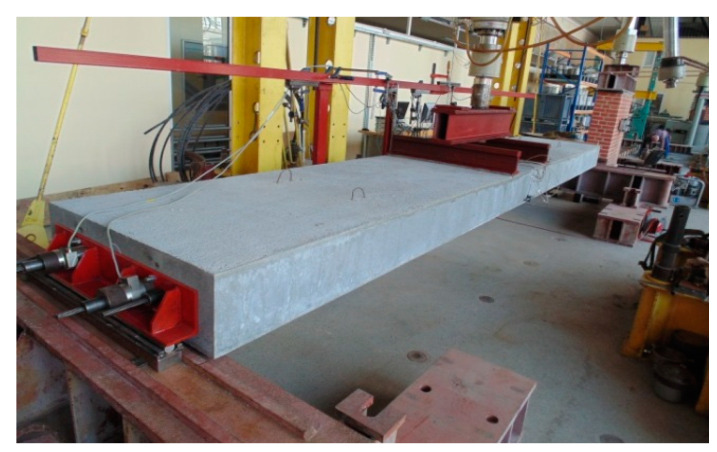
View of tested slab and load stand.

**Figure 10 materials-17-04154-f010:**
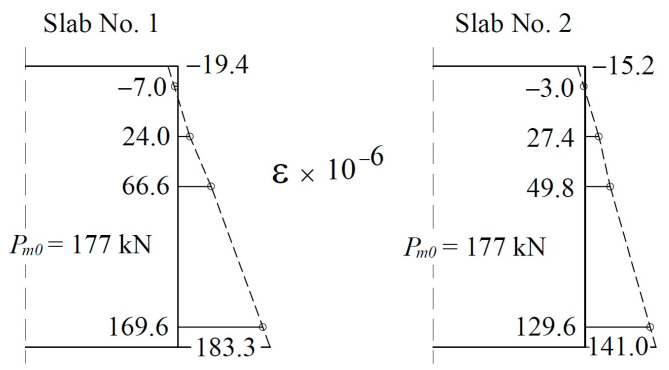
Middle span slab cross-section strains due to post-tensioning.

**Figure 11 materials-17-04154-f011:**
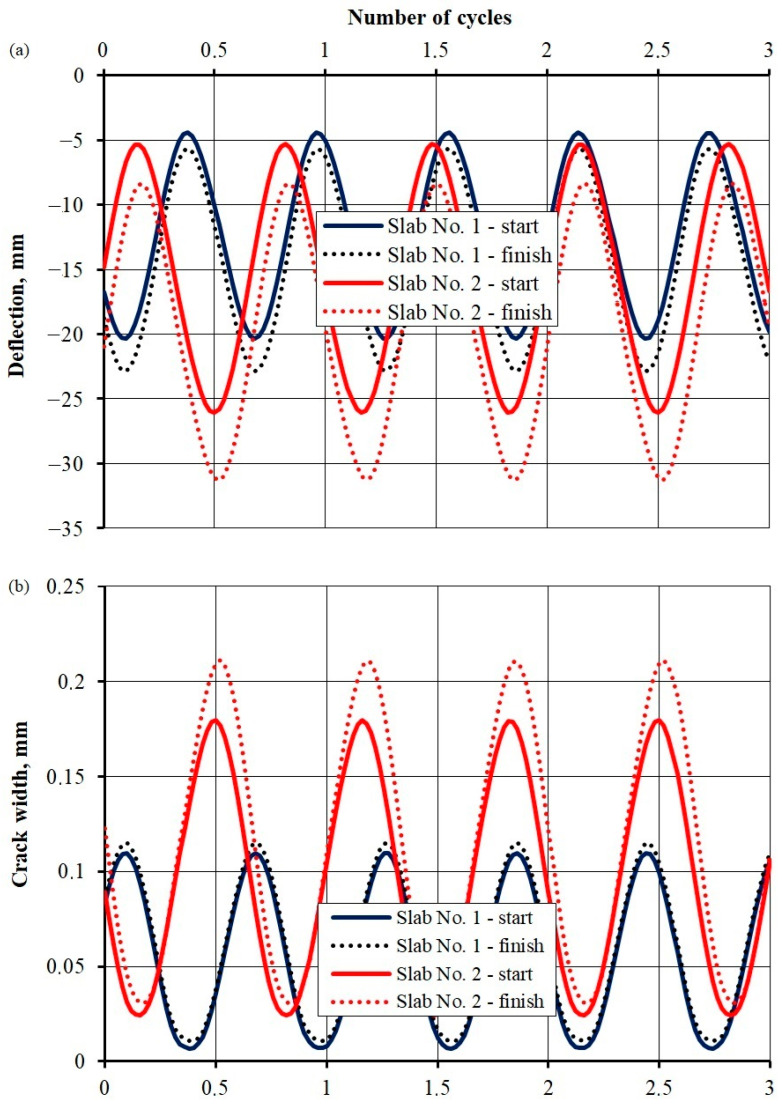
Deflection (**a**) and crack width course (**b**) during the beginning and end of the repetitive load test.

**Figure 12 materials-17-04154-f012:**
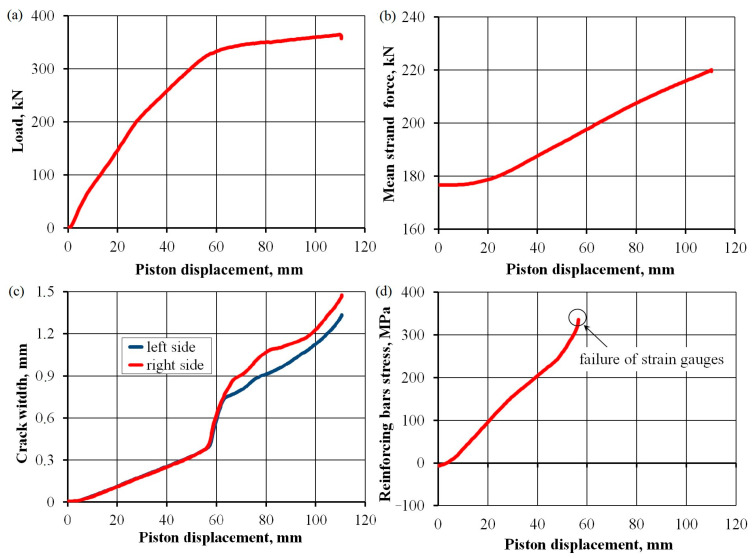
Development of selected parameters during the destruction test of slab No. 1: piston load value (**a**), mean strand force (**b**), crack width (**c**), and reinforcing bars stress (**d**).

**Figure 13 materials-17-04154-f013:**
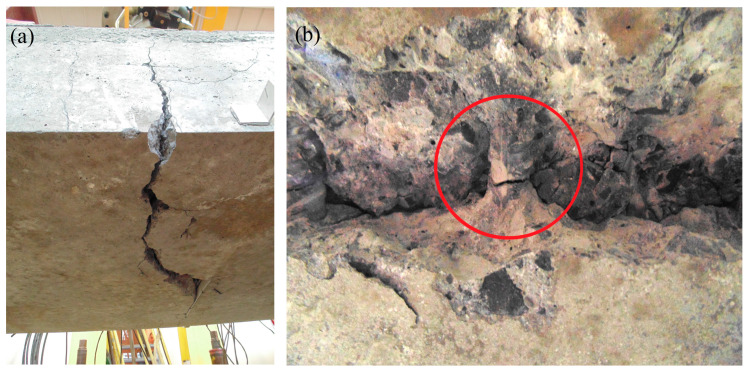
Broken slab No. 2: middle cross-section rupture (**a**); broken reinforcing bar (**b**) (red circle indicates a broken steel bar).

**Figure 14 materials-17-04154-f014:**
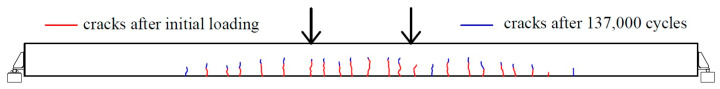
The cracks’ arrangement on the side face of slab No. 2 under a load of 180 kN.

**Figure 15 materials-17-04154-f015:**
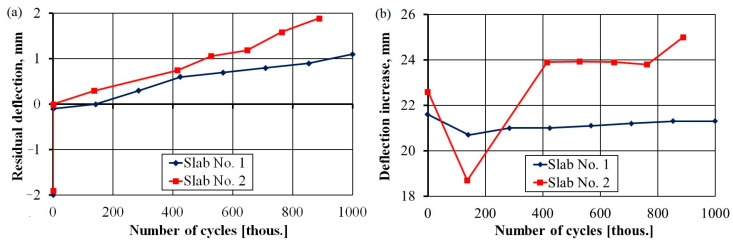
Residual deflection (**a**); growth of deflection under the control load (**b**) versus number of load cycles.

**Figure 16 materials-17-04154-f016:**
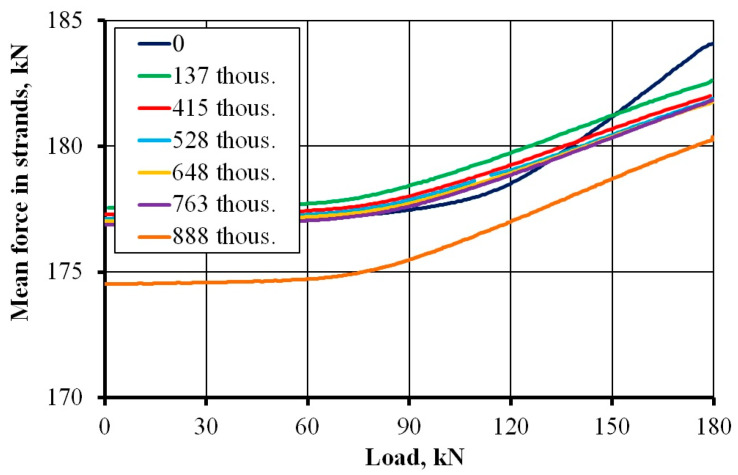
Mean force in strands versus load cycles in slab No. 2.

**Figure 17 materials-17-04154-f017:**
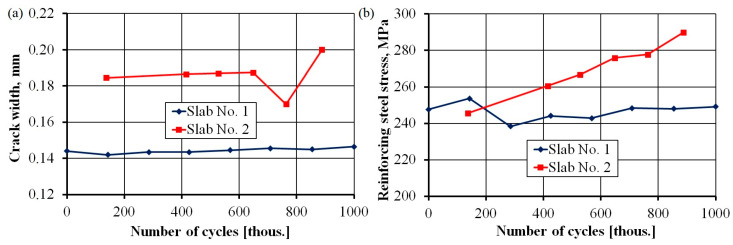
Crack width (**a**); reinforcing steel stress (**b**) versus number of load cycles.

**Table 1 materials-17-04154-t001:** Predicted values of the bending moment and stresses in the middle-span cross-section (for the load scheme presented in Figure 3).

Load	Bending Moment kNm	Upper Stress MPa	Bottom Stress MPa
1. Self-weight	28.4	2.1	−2.1
2. Prestressing after immediate prestress losses (*P_m0_* = 169.6 kN)	−38.4	−2.6	8.0
3. Prestressing after time-dependent prestress losses (*P_mt_* = 153 kN)	−34.6	−2.3	7.2
4. Vehicle (3 × 115 kN)	192.3	14.2	−14.2
Initial design situation (1 + 2)	-	−0.5	5.9
Final design situation (1 + 3 + 4)	-	14.0	−9.1

**Table 2 materials-17-04154-t002:** The results of the concrete’s mechanical properties 28 days after casting.

Properties	Mean Value MPa	Variation Coefficient *
Compressive strength	45.0	0.006
Modulus of elasticity	41,800	0.032
Splitting tensile strength	4.11	0.064
Modulus of rupture	7.38	0.040

* standard deviation/arithmetic average ratio.

## Data Availability

All research data will be made available after contacting the authors.

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
