# Peer review of "Post-Tensioned Hollow-Core Concrete Slabs with Unbonded Tendons for Truck Scale Platforms: Design Assumptions and Tests"

_materials, 2024, doi:10.3390/ma17164154_

Round 1

Reviewer 1 Report

Comments and Suggestions for Authors

Lines 46-51 - what were the findings of the studies cited in this paragraphs? How does your manuscript falls within the general state of the art?

Line 73 - the authors mention an axle force / load of 120kN but in all subsequent references the value of 115kN is mentioned. Which is the correct value?

Line 117 - the "overhang" the authors refer to is not indicated in Figure 5.

Line 146 - the sign convention used by the authors for tensile and compressive stresses is a little bit unusual. + sign is used for compressive stresses and - (minus) sign for tensile stresses. This makes understanding the paper very difficult due to the generally valid rule that stated quite the opposite in terms of sign convention.

Table 1 - please add to the caption that the loading diagram was the one presented in Figure 3. Please explain why there are 2 different values of the pre-stressing force presented in the table? Was the loss of prestress also taken into account?

Table 2 - please double check the unit of measure for the modulus of elasticity.

Line 195 - why were the slabs tested at 89 and 112 days? Moreover, tests on concrete samples should have been conducted at the 2 selected ages (in addition to the standard testing age of 28 days) in order to obtain the values of the mechanical properties at those considered ages.

Line 205 - unless they were cast from different batches or the values were obtained at different ages, the modulus of elasticity should not be significantly different in the 2 scenarios (taking into account that this elastic property is determined on the linear elastic range of material behavior). Please rethink this statement.

Line 209 - why are the values from Table 1 so different from the values that were actually measured and subsequently computed? Where could such a large difference come from?

Line 212 - how about the curing age and conditions of the two slabs?

Line 229 - what was the reason for selecting the load intensity of 180 kN?

Line 233 - why were 2 parameters changed at the same time: cycling load limits and frequency?

Section 4 - Please insert the figures as close as possible to their first reference in the text. The authors refer to multiple figures at the same time and the line of reasoning the difficult to follow.

Please redesign this section entirely.

Line 339 - there is another Section 4, titled Discussion but the content is hardly a discussion and interpretation of the obtained results. Please group the 2 sections 4 together and correlate your observed phenomena with previously reported results in the scientific literature.

Section 5 - please present the conclusions of the paper and not generalities not connected to the purpose of the manuscript. The second Section 4 above is more suitable for Conclusions.

Fig. 11 - decimal separator

Comments on the Quality of English Language

Line 27 - what do you mean by "are exclude each other ..."?

Line 31 - "Therefore, their .... becomes more expensive"

Line 34 - "while the weight of the platform..."

Line 35 - "the solution that fulfills all the requirements..."

Line 58 - what do you understand by "the vitality of a given slab"?

Line 86 - "large eccentric" or "large eccentricity"? What do you mean by "overhang of the tendons"?

Line 129 - please substitute "crossed" by "exceeded"

Line 136 - what "cores" are the authors referring to? The PVC pipes? Please rephrase for a better understanding.

Line 155 - what is the meaning of "dense cross-section"?

Line 203 - decimal separator: 169.6

Line 223 - was taken into account

Line 224 - the statement is not clear, please rephrase.

Author Response

We are grateful to the reviewer for his insightful analysis of the work.

All suggestions have been implemented.

I clarify the doubts below.

Comments 1: Lines 46-51 - what were the findings of the studies cited in this paragraphs? How does your manuscript falls within the general state of the art?

Response 1: The cited examples of work involved the use of unbonded tendons in columns and walls subjected to cyclic loads. These are completely different structures working under different conditions. The study of slabs working in the cracked condition presented by the authors is the first study of such elements, and no similar study has been carried out to date. Hence, we considered it pointless to report the results of tests of completely different structures. It is not possible to compare these results.

Comments 2: Line 73 - the authors mention an axle force / load of 120kN but in all subsequent references the value of 115kN is mentioned. Which is the correct value?

Response 2: The value 115 kN is correct. It has been corrected.

Comments 3: Line 117 - the "overhang" the authors refer to is not indicated in Figure 5.

Response 3: It has been added in Figure 5b

Comments 4: Line 146 - the sign convention used by the authors for tensile and compressive stresses is a little bit unusual. + sign is used for compressive stresses and - (minus) sign for tensile stresses. This makes understanding the paper very difficult due to the generally valid rule that stated quite the opposite in terms of sign convention.

Response 4: In the analysis and the design of concrete structures, we use the opposite convention than in structural mechanics. This is because the tensile stresses are undesirable for concrete, hence we mark them with “-”.

Comments 5: Table 1 - please add to the caption that the loading diagram was the one presented in Figure 3. Please explain why there are 2 different values of the pre-stressing force presented in the table? Was the loss of prestress also taken into account?

Response 5: Two values of prestress force are include the immediate and time-dependent prestress losses. The  explanations have been added to the Table 1.

Comments 6: Table 2 - please double check the unit of measure for the modulus of elasticity.

Response 6: It has been corrected.

Comments 7: Line 195 - why were the slabs tested at 89 and 112 days? Moreover, tests on concrete samples should have been conducted at the 2 selected ages (in addition to the standard testing age of 28 days) in order to obtain the values of the mechanical properties at those considered ages.

Response 7: The tests were conducted at two different times due to the duration of the test and the time taken to prepare the test stand. The slabs were tested on the same loading machine, loading the first slab took more than 10 days. The authors are aware that the specimens should have been tested at the time of loading each plate, but they no longer have the ability to turn back time and correct this test.

Comment 8: Line 205 - unless they were cast from different batches or the values were obtained at different ages, the modulus of elasticity should not be significantly different in the 2 scenarios (taking into account that this elastic property is determined on the linear elastic range of material behavior). Please rethink this statement.

Response 8: The authors had in mind the difference in modulus due to the effect of scale (small samples and full-scale structure). This was clarified.

Comments 9: Line 209 - why are the values from Table 1 so different from the values that were actually measured and subsequently computed? Where could such a large difference come from?

Response 9: The stress values given in Table 1 are design values, based on expected prestressed losses. These are always subject to some error. In addition, the stress values were measured indirectly by measuring the strains. This is not an accurate way, due to the need to take the modulus of elasticity in the sfull-scale tructure, based on the modulus tested on a small sample (as has been written about before). The randomness and difficult-to-predict behavior of concrete structures is well known. All the factors mentioned cause differences in predicted and measured values.

Comments 10: Line 212 - how about the curing age and conditions of the two slabs?

Response 10: Modulus of elsticity is a mechanical feature that grows the fastest of all. Thus, according to the authors, the difference in the age of the tested slabs had a negligible impact on the results of the obtained stresses.

Comments 11: Line 229 - what was the reason for selecting the load intensity of 180 kN?

Response 11: The force of 180 kN was selected so as to obtain the same value of bending moment with a two-point load as with a three-point load of 3x60 kN (taking into account a 10% of overload of the slab).

Comment 12: Line 233 - why were 2 parameters changed at the same time: cycling load limits and frequency?

Response 12: The research was aimed at testing boards used in industry. Since the first plate did not deteriorate after the cyclic load, for the second plate the load value was increased to test its behavior under a higher load. The machine's performance, however, required a reduction in frequency.

Comments 13: Section 4 - Please insert the figures as close as possible to their first reference in the text. The authors refer to multiple figures at the same time and the line of reasoning the difficult to follow. Please redesign this section entirely.

Response 13: It has been improved.

Comments 14:

Line 339 - there is another Section 4, titled Discussion but the content is hardly a discussion and interpretation of the obtained results. Please group the 2 sections 4 together and correlate your observed phenomena with previously reported results in the scientific literature.

Comments 15:

Section 5 - please present the conclusions of the paper and not generalities not connected to the purpose of the manuscript. The second Section 4 above is more suitable for Conclusions.

Response 14 and 15:

The doubled sections 4 have been removed and its content has been moved to conclusions. It is impossible to compare the obtained results with the scientific literaturę. Similar studies have not been done and published. Such a comment was made in the conclusions.

Comments 16: Fig. 11 - decimal separator

Response 16: It has been corrected.

Comments on the Quality of English Language

Response: All corrections have been made. The word „cores” has been maintained. „Hollow-cores”  means empty ducts, regardless of how they were formed.

„Dense cross-section” is used to denote a fully filled section, with no voids.

Reviewer 2 Report

Comments and Suggestions for Authors

I had the opportunity to read 'Post-tensioned hollow-core concrete slabs with unbounded tendons for truck scale platforms. Design assumptions and tests' manuscript submitted for publication to Materials journal.

The manuscript is valuable and deserves publication.

Here are my comments:

- first two paragraphs of introduction must be supported by references. There are none.

- bulk references must be avoided ("[2=7]", l. 46).

- '2. Materials and Methods' section is in need of references too.

- software used to do the drawings should be mentioned.

- not all people are familiarized with the terminology. Take for instance variation coefficient. A formal definition is welcomed.

- a typo error in line 209 should be corrected ("2.6 i 8.0").

- discussions should be conducted with support from scientific literature - references to similar studies with different and/or similar results.

- it is  clear for me that the cracks from the slab are influenced by the composition and structure of the material; I would like to see the opinion of the authors on the regard of use of an alternate hypothetic ultra-elastic material such as is all carbon made triple crossed C28 cyclic polyyne cluster. if it is possible, please include it into the discussion.

- please enrich your literature survey; use journal published papers.

- please also discuss the extent in which this case study can be used to generate a more general methodology.

Author Response

We would like to thank the reviewer for studying the work and for his valuable comments. I have made changes as far as I could. Below are the explanations.

Comments 1: First two paragraphs of introduction must be supported by references. There are none.

Response 1: Two items available in English have been introduced. This is extremely difficult because there is a lack of scientific publications in this area.

Comments 2: Bulk references must be avoided ("[2=7]", l. 46).

Response 2: This has been separated.

Comments 3: 2. Materials and Methods' section is in need of references too.

Response 3: This is the author's chapter. The research was designed and performed based on the author's own knowledge and experience. Such research has not been performed and published so far, hence it is difficult to cite publications in this area.

Comments 4: Software used to do the drawings should be mentioned.

Response 4: The analysis of bending moments and cross-sections stresses was performer on the one-span simply supported element.  These are simply calculations possibly to make withouh computer software.

Comments 5: Not all people are familiarized with the terminology. Take for instance variation coefficient. A formal definition is welcomed.

Response 5: The explanation has been added bellow Table 2

Comments 6: A typo error in line 209 should be corrected ("2.6 i 8.0").

Response 6: It has been corrected.

Comments 7: Discussions should be conducted with support from scientific literature - references to similar studies with different and/or similar results.

Response 7: The solution shown is probably the only one of its kind in the world. Post-tensioned  slabs with unbounded tendons, operating in the cracked state under cyclic loading, have not been tested before. Thus, there is impossible to compare obtained results with the results of similat test.

Comments 8: It is  clear for me that the cracks from the slab are influenced by the composition and structure of the material; I would like to see the opinion of the authors on the regard of use of an alternate hypothetic ultra-elastic material such as is all carbon made triple crossed C28 cyclic polyyne cluster. if it is possible, please include it into the discussion.

Response 8: The concrete is commonly used material in this type of structures. Cracking in concrete, caused by lack of its ductility, is common and accepted phenomena. The authors didn’t consither the other composits, thus no mention of them in the discussion.

Comments 9: Please enrich your literature survey; use journal published papers.

Response 9: It is difficult to find publications in this area, as similar research has not been done so far.

Comments 10: Please also discuss the extent in which this case study can be used to generate a more general methodology.

Response 10: It has been added to conclusions.

Round 2

Reviewer 1 Report

Comments and Suggestions for Authors

Thank you for taking into account eh suggestions made during the reviewing process.

In my opinion, the manuscript fulfills the quality adn scientific soundness requirements to be published.